# Analytical Solution of the Susceptible-Infected-Recovered/ Removed Model for the Not-Too-Late Temporal Evolution of Epidemics for General Time-Dependent Recovery and Infection Rates

Reinhard Schlickeiser [1,2,*] and Martin Kröger [3,*]

1   Institut für Theoretische Physik, Lehrstuhl IV: Weltraum- und Astrophysik, Ruhr-Universität Bochum,
    D-44780 Bochum, Germany
2   Institut für Theoretische Physik und Astrophysik, Christian-Albrechts-Universität zu Kiel, Leibnizstr. 15,
    D-24118 Kiel, Germany
3   Magnetism and Interface Physics & Computational Polymer Physics, Department of Materials, ETH Zurich,
    CH-8093 Zurich, Switzerland
*   Correspondence: rsch@tp4.rub.de (R.S.); mk@mat.ethz.ch (M.K.)

**Abstract:** The dynamical equations of the susceptible-infected-recovered/removed (SIR) epidemics model play an important role in predicting and/or analyzing the temporal evolution of epidemic outbreaks. Crucial input quantities are the time-dependent infection ($a(t)$) and recovery ($\mu(t)$) rates regulating the transitions between the compartments $S \to I$ and $I \to R$, respectively. Accurate analytical approximations for the temporal dependence of the rate of new infections $\mathring{J}(t) = a(t)S(t)I(t)$ and the corresponding cumulative fraction of new infections $J(t) = J(t_0) + \int_{t_0}^{t} dx \mathring{J}(x)$ are available in the literature for either stationary infection and recovery rates or for a stationary value of the ratio $k(t) = \mu(t)/a(t)$. Here, a new and original accurate analytical approximation is derived for general, arbitrary, and different temporal dependencies of the infection and recovery rates, which is valid for not-too-late times after the start of the infection when the cumulative fraction $J(t) \ll 1$ is much less than unity. The comparison of the analytical approximation with the exact numerical solution of the SIR equations for different illustrative examples proves the accuracy of the analytical approach.

**Keywords:** epidemics; temporal development; coronavirus; SARS CoV-2; COVID-19



## 1. Introduction

The susceptible-infected-recovered/removed (SIR) epidemics model, originally developed by Kermack and McKendrick [1] and refined by Kendall [2], is the simplest realistic and, therefore, often-applied description of the temporal evolution of epidemics [3–47]. Here, persons from the considered population are assigned to the three compartment fractions $S$ (susceptible), $I$ (infectious), and $R$ (recovered/removed). The time-dependent infection ($a(t)$) and recovery ($\mu(t)$) rates regulate the transitions between the compartments $S \to I$ and $I \to R$, respectively. For a review, see [48].

In general, the time dependencies of these two rates will be different and determined by different factors: the dedicated medication of infected persons will increase the recovery rate from its initial value at the start of an epidemic outbreak, while nonpharmaceutical interventions, such as social distancing, quarantine, and mask obligations, effectively reduce the infection rate from its initial value. As a consequence, the ratio $k(t) = \mu(t)/a(t)$ of the two rates is a time-dependent function, thereby increasing at early times from its initial value and decreasing at later times when some of the nonpharmaceutical interventions are lifted. Such a behavior of the ratio $k(t)$ has indeed been established recently from the analyis of past COVID-19 mutants [49], as the ratio $k(t)$ can be expressed in terms of the well-monitored rate of new infections $\mathring{J}(t) = dJ(t)/dt$ and its corresponding cumulative

fraction $J(t)$. In order to improve the forecast of future epidemic outbreaks using the nonstationary ratio $k(t)$, it is, therefore, highly desirable to derive analytical solutions or accurate approximations of these solutions of the SIR model equations for arbitrary but given time dependencies of the infection and recovery rates and their ratios defined by $k(t)$. This is the purpose of the present manuscript.

In the literature, very often the SIR model equations have been solved numerically with adopted stationary infection ($a_0$) and recovery ($\mu_0$) rates so that their ratio $k_0 = \mu_0/a_0$ is also stationary, although an analytical solution in terms of an inverse integral in this case is available [3]. Additionally, analytical solutions for arbitrary but given time dependencies of the infection rate $a(t)$ have been derived for the infinite [4] and semitime ($t \geq t_0$) [5] time domains for the case of a stationary ratio $k = \mu(t)/a(t)$, thereby implying that the recovery rate has exactly the same time dependence as the infection rate. Analytical approximations have been developed [50] for slowly varying ratios of $k(t)$ in comparison with the typical time characteristics of the epidemic wave. Below, we will derive approximate analytical solutions of the SIR model equations for the limit of not-too-late times, where the cumulative number of new infections $J$ is much smaller than unity. For completeness, we investigate the alternative, but less interesting, case at late times, when the fraction of susceptible persons $S \ll 1$ is much smaller than unity (Appendix B).

## 2. SIR Model

The original SIR equations for the three-compartment fractions $S(t)$, $I(t)$, and $R(t)$ at given time-dependent infection and recovery rates read as follows [1,3,48]:

$$\frac{dS}{dt} = -a(t)SI, \tag{1a}$$

$$\frac{dI}{dt} = a(t)SI - \mu(t)I, \tag{1b}$$

$$\frac{dR}{dt} = \mu(t)I, \tag{1c}$$

where obeying the sum constraint yields

$$S(t) + I(t) + R(t) = 1 \tag{2}$$

at all semitimes $t \geq t_0$ after the start of the wave at time $t_0$, which are subject to the initial conditions [5]

$$I(t_0) = \eta, \qquad S(t_0) = 1 - \eta, \qquad R(t_0) = 0, \tag{3}$$

where $\eta$ is positive and usually very small such that $\eta \ll 1$. In terms of the reduced time, we have the following:

$$\tau \equiv \int_{t_0}^{t} dx\, a(x) \tag{4}$$

so that $\tau = 0$ at $t = t_0$, and the ratio is defined as follows:

$$k(\tau) \equiv \frac{\mu(\tau(t))}{a(\tau(t))} \tag{5}$$

Thus, the SIR set of equations in (1) for $S(\tau)$, $I(\tau)$, and $R(\tau)$ read as follows [4]:

$$\frac{dS}{d\tau} = -SI, \tag{6a}$$

$$\frac{dI}{d\tau} = SI - k(\tau)I, \tag{6b}$$

$$\frac{dR}{d\tau} = k(\tau)I. \tag{6c}$$

Recently, it has been demonstrated [49] that the reduced SIR set of equations in (6) is equivalent to

$$k(\tau) \quad = \quad 1 - J(\tau) - \frac{d}{d\tau} \ln \left[ \frac{d}{d\tau} \ln(1 - J(\tau))^{-1} \right], \tag{7}$$

where

$$J(\tau) = J(0) + \int_0^\tau d\xi \, j(\xi) = 1 - S(\tau) \tag{8}$$

denotes the cumulative number of new infections, and

$$j(\tau) = S(\tau)I(\tau) = -\frac{dS(\tau)}{d\tau} \tag{9}$$

denotes the rate of new infections. Using Equation (8), we can also write Equation (7) as

$$
\begin{aligned}
k(\tau) \quad &= \quad S(\tau) - \frac{d}{d\tau} \ln \left[ \frac{d}{d\tau} \ln \frac{1}{S(\tau)} \right] \\
&= \quad S(\tau) - \frac{d}{d\tau} \ln I(\tau),
\end{aligned}
\tag{10}
$$

where we used Equation (9) as well for the revision. In the following, we will derive approximate analytical solutions of the two nonlinear differential Equations (7) and (10) in the two limits of small $J(\tau) \ll 1$ and large $J(\tau) \simeq J_\infty$, respectively.

The first limit $J \ll 1$ holds at early reduced times $\tau \le \tau_c$ of the epidemic outbreak and corresponds to values of $S$ that are smaller but very close to $S(t_0)$. Provided it is reached, which depends on the reduced time dependence of the ratio $k(\tau)$, the second limit $J \simeq J_\infty$ holds at the late reduced time $\tau > \tau_c$. As a rule of thumb [51], any pandemic wave ends when 70% of the total population are infected, i.e. $J_\infty = 0.7$, if nothing is done to reduce the number of infections. We investigate the early time limit next. For completeness, we study the less interesting late time limit in Appendix B.

## 3. Approximate Analytical Solutions
### 3.1. Solution in the Limit of Small $J \ll 1$

Initially at a reduced time $\tau = 0$, the cumulative number of new infections is extremely small, where $J(0) = \eta$. In the limit $J(\tau) \ll 1$ and at later times where $0 \le \tau \le \tau_c$, we use the approximations $1 - J(\tau) \simeq 1$ and $\ln(1 - J(\tau))^{-1} \simeq J(\tau)$ to obtain the ratio derived from (7):

$$k(\tau) \simeq 1 - \frac{d}{d\tau} \ln \left[ \frac{dJ(\tau)}{d\tau} \right] = 1 - \frac{d}{d\tau} \ln j(\tau), \tag{11}$$

which immediately integrates to

$$j(\tau \le \tau_c) = \eta(1 - \eta)e^{\tau - \int_0^\tau d\xi \, k(\xi)} = \eta(1 - \eta)e^{\int_0^\tau d\xi \, [1 - k(\xi)]}, \tag{12}$$

where we make use of the initial condition $j(0) = \eta(1 - \eta)$. A further integration of (12) provides

$$J(\tau \le \tau_c) = \eta + \eta(1 - \eta) \int_0^\tau d\tau' e^{\tau' - \int_0^{\tau'} d\xi \, k(\xi)}. \tag{13}$$

In Appendix A, the integral of (13) in the case of general variations in $k(\tau)$ is evaluated with the method of steepest descent in terms of error functions as

$$J(\tau \leq \tau_c) \simeq \eta + \eta(1-\eta) \sum_m \sqrt{\frac{\pi}{2k'(\tau_m)}} \exp\left(\tau_m - \int_0^{\tau_m} d\xi\, k(\xi)\right) \times$$
$$\left[\mathrm{erf}\left(\sqrt{\frac{k'(\tau_m)}{2}}(\tau - \tau_m)\right) + \mathrm{erf}\left(\sqrt{\frac{k'(\tau_m)}{2}}\tau_m\right)\right], \tag{14}$$

where $k'(\tau)$ denotes $dk(\tau)/d\tau$. In terms of the real time in this early time limit, one has

$$\mathring{J}(t \leq t_c) = a(t)\eta(1-\eta)e^{\int_{t_0}^t dx[a(x)-\mu(x)]}. \tag{15}$$

*3.2. Properties of the Approximate Solution (12)*

The approximate solution (12) is predominantly determined by the reduced time variation of the ratio $k(\tau)$. For the first and second time derivatives of the solution (12), we obtain

$$\frac{dj}{d\tau} = \eta(1-\eta)[1-k(\tau)]e^{\tau - \int_0^\tau d\xi\, k(\xi)}, \tag{16}$$

$$\frac{d^2j}{d\tau^2} = \eta(1-\eta)\left[(1-k(\tau))^2 - k'(\tau)\right]e^{\tau - \int_0^\tau d\xi\, k(\xi)}. \tag{17}$$

Consequently, the extrema of the rate of new infections occur at reduced times $\tau_E$, which are determined by

$$k(\tau_E) = 1. \tag{18}$$

Then,

$$\left.\frac{d^2j}{d\tau^2}\right|_{\tau=\tau_E} = -\eta(1-\eta)k'(\tau_E)e^{\tau_E - \int_0^{\tau_E} d\xi k(\xi)}, \tag{19}$$

thus meaning that the extrema are maxima for an increasing reduced time variation, with $k'(\tau_E) > 0$, and are minima for a decreasing reduced time variation, with $k'(\tau_E) < 0$. The extreme values of the rate of new infections are given by

$$j_E(\tau_E) = \eta(1-\eta)e^{\int_0^{\tau_E} d\xi\, [1-k(\xi)]}. \tag{20}$$

**4. Special Cases**

*4.1. Constant Ratio $k(t)$*

We first consider the special case of a stationary ratio $k(t) = k_0 =$const. for which very accurate analytical approximations have been derived [5,52]. In this case, the approximations (12)–(15) reduce to

$$j(\tau \leq \tau_c) = \eta(1-\eta)e^{(1-k_0)\tau}, \tag{21a}$$

$$J(\tau \leq \tau_c) = \eta + \frac{\eta(1-\eta)}{1-k_0}\left[e^{(1-k_0)\tau} - 1\right], \tag{21b}$$

$$\mathring{J}(t \leq t_c) = \frac{a(t)\eta(1-\eta)}{1-k_0}e^{(1-k_0)\int_{t_0}^t dx\, a(x)}, \tag{21c}$$

and determining Equation (A15) becomes

$$e^{(1-k_0)\tau_c} - 1 = \frac{J_\infty(1-k_0)}{\eta(1-\eta)} = \frac{0.7(1-k_0)}{\eta(1-\eta)}, \tag{22}$$

or, equivalently,

$$\tau_c = \frac{1}{1-k_0} \ln\left[1 + \frac{0.7(1-k_0)}{\eta(1-\eta)}\right] \simeq \frac{\ln\frac{0.7(1-k_0)}{\eta}}{1-k_0}, \tag{23}$$

which agrees favorably well with the exact numerical result for the time $\tau_c$ at which $J(t)$ has reached half of its final valu $J_\infty$. (see Figure 1).

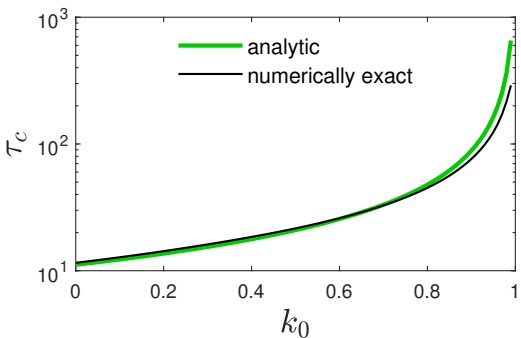

**Figure 1.** Performance of the analytic approximation (23) for the crossover $\tau_c$ versus $k_0$ at $\eta = 10^{-5}$. The numerical result is obtained from $J(\tau_c) = J_\infty/2$.

*4.2. Linearly Increasing Ratio $k(\tau) = k_0 + k_1\tau$*

We note that for a linearly increasing ratio $k(\tau) = k_0 + k_1\tau$, the analytical approximation (12) is given by the Gaussian distribution

$$j(\tau \leq \tau_c) = \eta(1-\eta)e^{-\left[\frac{1}{2}k_1\tau^2 - (1-k_0)\tau\right]}, \tag{24}$$

which, for a constant infection rate $a_0$ yielding $\tau = a_0(t - t_0)$, leads to a Gaussian distribution in real time. Such Gaussian distributions have been successfully used to predict the temporal evolution of earlier COVID-19 waves [51,53–56].

## 5. Illustrative Examples

In order to illustrate the usefulness of our approximate solution (12), we consider two illustrative examples for the reduced time variation of the ratio $k(\tau)$. The first one is monotonically rising and therefore well suited to represent a single wave of a pandemic outburst. The second one varies periodically in reduced time and is therefore well suited to represent a series of repeating pandemic outbursts. In both cases, we compare the exact numerical solution of the SIR set of equations in (6) with the approximative solution (12). We consider both examples in turn.

*5.1. Monotonically Rising Ratio*

Here, we choose

$$k(\tau) = B\tanh(C\tau), \tag{25}$$

with the two positive constants being $B$ and $C$. The ratio (25) increases monotonically from zero at $\tau = 0$ to its maximum value $B$ at large reduced times of $\tau \gg B^{-1}$. According to Equation (18), a single extremum of the rate of new infections occurs at the time $\tau_E$ and is given by

$$B\tanh(C\tau_E) = 1, \tag{26}$$

which can only be solved for values of $B \geq 1$ with

$$\tau_E = C^{-1}\text{arctanh}\left(B^{-1}\right) = \frac{1}{C}\ln\frac{B+1}{\sqrt{B^2-1}}. \tag{27}$$

Because the first derivative of the ratio (25) is given by

$$\frac{dk}{d\tau} = \frac{BC}{\cosh^2(C\tau)} = \left[1 - \tanh^2(C\tau)\right]BC, \tag{28}$$

one finds

$$k'(\tau_E) = \frac{C(B^2 - 1)}{B} > 0, \tag{29}$$

thus meaning that the extremum is a maximum. For values of $B < 1$, the rate of new infections monotonically increases with reduced time. With the choice of (25), the rate of new infections of (12) can be reduced to

$$
\begin{aligned}
j(\tau \leq \tau_c) &= \eta(1 - \eta)e^\tau[\cosh(C\tau)]^{-B/C} \\
&= \eta(1 - \eta)e^\tau[1 - \tanh^2(C\tau)]^{B/2C}.
\end{aligned} \tag{30}
$$

We note the two asymptotic exponential behaviors $j(\tau \leq C^{-1} \leq \tau_c) \simeq \eta(1 - \eta)e^\tau$ and $j(C^{-1} \leq \tau \leq \tau_c) \simeq \eta(1 - \eta)2^{B/C}e^{(1-B)\tau}$.

Only for values of $B > 1$, a single maximum rate

$$
\begin{aligned}
j_{\max} = j(\tau_E) &= \eta(1 - \eta)\left[\frac{B^2 - 1}{B^2}\right]^{\frac{B}{2C}}e^{C^{-1}\operatorname{arctanh}(1/B)} \\
&= \eta(1 - \eta)\frac{(B^2 - 1)^{\frac{B-1}{2C}}(B + 1)^{\frac{1}{C}}}{B^{\frac{B}{C}}}
\end{aligned} \tag{31}
$$

occurs at $\tau_E$, provided that $\tau_E \leq \tau_c$, or for $\tau_c$ in the case where $\tau_E > \tau_c$, we have

$$j_{\max} = j(\tau_c) = \eta(1 - \eta)e^{\tau_c}[\cosh(C\tau_c)]^{-\frac{B}{C}}. \tag{32}$$

In Figure 2 (center panels), we compare the approximative analytical rate of new infections of (30) as a function of the reduced time for this choice of the ratio $k(\tau)$ with the exact numerical solution of the SIR set of equations in (6). The agreement is almost perfect, thereby proving the accuracy of our analytical approximation.

The corresponding cumulative number of infections is given by

$$J(\tau) = J(0) + \int_0^\tau dx\, j(x) = \eta + \eta(1 - \eta)H(\tau), \tag{33}$$

$$H(\tau) = \int_0^\tau dx\, e^x[\cosh(Cx)]^{-B/C}. \tag{34}$$

By substituting $y = e^{2Cx}$, which corresponds to $x = \ln(y)/2C$, the integral (34) becomes

$$H(\tau) = \frac{2^{B/C}}{2C}\int_1^{e^{2C\tau}} dy\, y^{\frac{B+1}{2C} - 1}(1 + y)^{-B/C}, \tag{35}$$

which can be expressed as the difference of two hypergeometric $_2F_1$ functions by using integral 3.194 of [57]. One then obtains

$$
\begin{aligned}
H(\tau) = \frac{2^{B/C}}{B + 1}\Bigg[ &e^{(B+1)\tau}\,_2F_1\left(\frac{B}{C}, \frac{B+1}{2C}; 1 + \frac{B+1}{2C}; -e^{2C\tau}\right) \\
&- {}_2F_1\left(\frac{B}{C}, \frac{B+1}{2C}; 1 + \frac{B+1}{2C}; -1\right)\Bigg].
\end{aligned} \tag{36}
$$

For $\tau \ll 1$, the first terms of the series expansion are $H(\tau) = \tau + \tau^2/2 + (1 - BC)\tau^3/6$, which are derived in Appendix C.

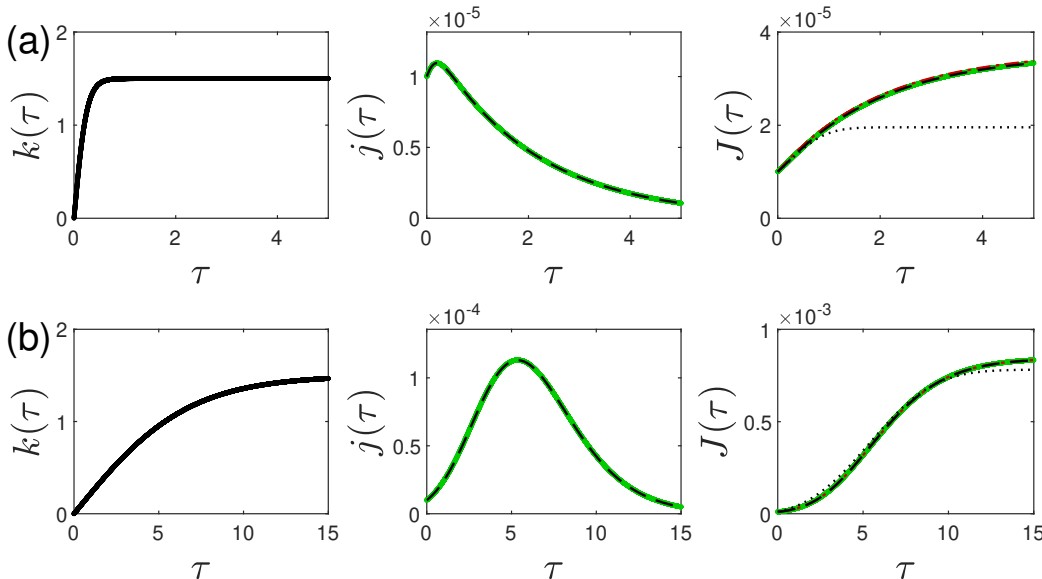

**Figure 2.** Left: Plot of the monotonically rising ratio (25) for (**a**) $C = 4$ and (**b**) $C = 0.15$, with $B = 1.5$ and $\eta = 10^{-5}$, respectively. Middle: Corresponding rate of new infections $j(\tau)$ as a function of the reduced time for these choices of the ratio $k(\tau)$. Right: Corresponding cumulative rate of new infections $J(\tau)$. Shown are the numerical (black dashed curve) solutions of the SIR set of equations in (6) in comparison with the analytical approximations (green curve) according to Equations (12) and (30). The agreement is almost perfect. The maximum relative deviations are smaller than (**a**) 0.5% and (**b**) 2.5%. In addition, we show (hardly visible red dot–dash curve) the approximant (39) in panel (**a**) for $C > 1$ and Equation (36) in panel (**b**) for $C < 1$. The agreement is against almost perfect. The black dotted lines were obtained for comparison using the method of steepest descent, Equation (45), which provides a lower limit to the cumulative fraction, because all other contributions far from the maximum are not adequately accounted for (Appendix A).

For $C > 1$, an approximant can be derived by substituting $x = \ln y$ in Equation (34). This yields

$$
\begin{aligned}
H(\tau) &= 2^{B/C} \int_1^{e^\tau} dy \, [y^C + y^{-C}]^{-B/C} \\
&= 2^{B/C} \int_1^{e^\tau} dy \, y^B [1 + y^{2C}]^{-B/C}.
\end{aligned}
\tag{37}
$$

Since $y$ is greater than unity, we approximate $1 + y^{2C} \simeq y^{2C}$ to yield

$$
H(\tau) \simeq 2^{B/C} \int_1^{e^\tau} dy \, y^{-B} = \frac{e^{(1-B)\tau} - 1}{1 - B}.
\tag{38}
$$

Consequently, the cumulative fraction of infections of (33) becomes

$$
\begin{aligned}
J(\tau) &\simeq \eta + \eta(1-\eta) 2^{B/C} \frac{e^{(1-B)\tau} - 1}{1 - B} \\
&= \eta + \eta(1-\eta)
\begin{cases}
2^{B/C} \frac{e^{(1-B)\tau}-1}{1-B} & \text{for } B < 1 \\
2^{1/C} \tau & \text{for } B = 1 \\
2^{B/C} \frac{1 - e^{-(B-1)\tau}}{B-1} & \text{for } B > 1
\end{cases}
\end{aligned}
\tag{39}
$$

We first note that the absolute level of the cumulative fraction is proportional to $2^{B/C}$. If the parameter $C$ is small, one obtains a much higher amplification of the cumulative fraction at later times compared to its initial value than in cases where $C$ is large. This is clearly evident from the last panels of Figure 2.

Moreover, we notice that for values of $B > 1$, the cumulative fraction (39) approaches the finite value $J(\tau = \infty, B > 1) = \eta(1 - \eta)2^{B/C}/(B - 1)$, which is much smaller than $J_\infty = 0.7$, since $\eta \ll 1$. In this case, the early time solutions of (30) and (39) are valid for all times. This is easy to understand, because for values of $B > 1$, the ratio (25) becomes greater than unity after finite times so that then the recovery rate $\mu(\tau) > a(\tau)$ is greater than the infection rate; therefore, the rate of new infections is decreasing with time in agreement with the right side of Figure 2. In this case, not many new infections add to the cumulative fraction.

The opposite behavior holds for values of $B \leq 1$. In this case, at all times, the ratio (25) is less than or equal to unity so that the infection rate is never smaller than the recovery rate. Consequently, the cumulative fraction (39) increases exponentially with time for $B < 1$ and linearly with time for $B = 1$. In this case, the early time solutions of (30) and (39) can only be used for times less than $\tau_c$, which are provided according to Equation (A15), and we work with the following equation:

$$\int_0^{\tau_c} d\tau \, e^\tau [\cosh(C\tau)]^{-B/C} = H(\tau_c) = \frac{J_\infty}{\eta(1 - \eta)}. \tag{40}$$

Using Equation (39), we obtain

$$\tau_c = \frac{1}{1 - B} \ln\left[1 + \frac{(1 - B)J_\infty}{2^{B/C}\eta(1 - \eta)}\right]. \tag{41}$$

For $B \leq 1$, the rate of new infections (A12) at late times becomes

$$\begin{aligned} j(\tau \geq \tau_c) &\simeq (1 - J_\infty)A_1 \cosh^{-B/C}(C\tau) \\ &= \eta(1 - \eta)e^{\tau_c} \cosh^{-B/C}(C\tau), \end{aligned} \tag{42}$$

where we determined $A_1 = \eta(1 - \eta)e^{\tau_c}/(1 - J_\infty)$ from equating the two rates of (30) and (42) at $\tau_c$, whose value is given explicitly by Equation (41).

For the interesting case of values $B > 1$, we calculate the cumulative fraction of (33) with the integral of $H(\tau)$ using the method of steepest descent according to Equation (14). Here, a single maximum in $j(\tau)$ occurs at

$$\tau_m = \tau_E = \frac{1}{C} \ln \frac{B + 1}{\sqrt{B^2 - 1}}, \tag{43}$$

which is given by Equation (27) and $k'(\tau_m) = (B^2 - 1)C/B$, which is inferred from Equation (29). Moreover,

$$\begin{aligned} \tau_m - \int_0^{\tau_m} d\xi \, k(\xi) &= \tau_m - \frac{B}{C} \ln \cosh(C\tau_m) \\ &= \frac{1}{C}\left[\ln \frac{B + 1}{\sqrt{B^2 - 1}} - B \ln \frac{B}{\sqrt{B^2 - 1}}\right]. \end{aligned} \tag{44}$$

Consequently, the cumulative number of infections of (14) in this case becomes

$$\begin{aligned} J(\tau) \simeq \ &\eta + \eta(1 - \eta)\sqrt{\frac{\pi B}{2C}}(B + 1)^{\frac{1}{C}} B^{-\frac{B}{C}}(B^2 - 1)^{\frac{B-C-1}{2C}} \times \\ &\left[\mathrm{erf}\left(\sqrt{\frac{C\sqrt{B^2 - 1}}{2B}}(\tau - \tau_m)\right) + \mathrm{erf}\left(\sqrt{\frac{C\sqrt{B^2 - 1}}{2B}}\tau_m\right)\right]. \end{aligned} \tag{45}$$

In Figure 2 on the right, we compare the approximation of (45) with the exactly integrated cumulative fractions of (33)–(39) in this case. The good agreement indicates that the method of steepest descent is indeed appropriate to calculate cumulative fractions for $B > 1$ and

$C < 1$. For values of $C > 1$, this method is correct within a factor of 1.8, as indicated by Figure 2a (right). The reason that the method of steepest descent works better for small values of $C$ is due to the inverse dependence of the exponents in Equation (31) for the maximum rate and in Equation (27) for the time of maximum. Consequently, the smaller the value of C, the later the time that the maximum appears and with a larger amplitude as compared to the case of large values of $C$. In the case of a large amplitude maximum, the dominating contribution to the cumulative fraction stems from the maximum.

*5.2. Oscillating Ratio*

Here we choose

$$k(\tau) = 1 + \alpha \sin(\beta\tau), \tag{46}$$

with the two positive constants $\alpha$ and $\beta$. The ratio (46) oscillates periodically around its intial value of one (see Figure 3a). It is therefore well suited to represent a series of repeating pandemic outbursts. For values of $k$ that are greater than unity, the recovery rate is greater than the infection rate; so, the rate of infections decreases. When the ratio $k$ is smaller than unity, the infection rate is greater than the recovery rate, and the rate of infections increases with time.

With the choice of (46) and using the rate of new infections of (12) at early times, we have

$$j(\tau \le \tau_c) = \eta(1-\eta)e^{\frac{\alpha}{\beta}[\cos(\beta\tau)-1]}. \tag{47}$$

The approximative analytical rate of new infections (47) as a function of the reduced time for the oscillating ratio (46) is compared with the exact numerical solution of the SIR set of equations in (6) in Figure 3. The agreement is almost perfect, thereby proving the accuracy of our analytical approximation using Equation (12).

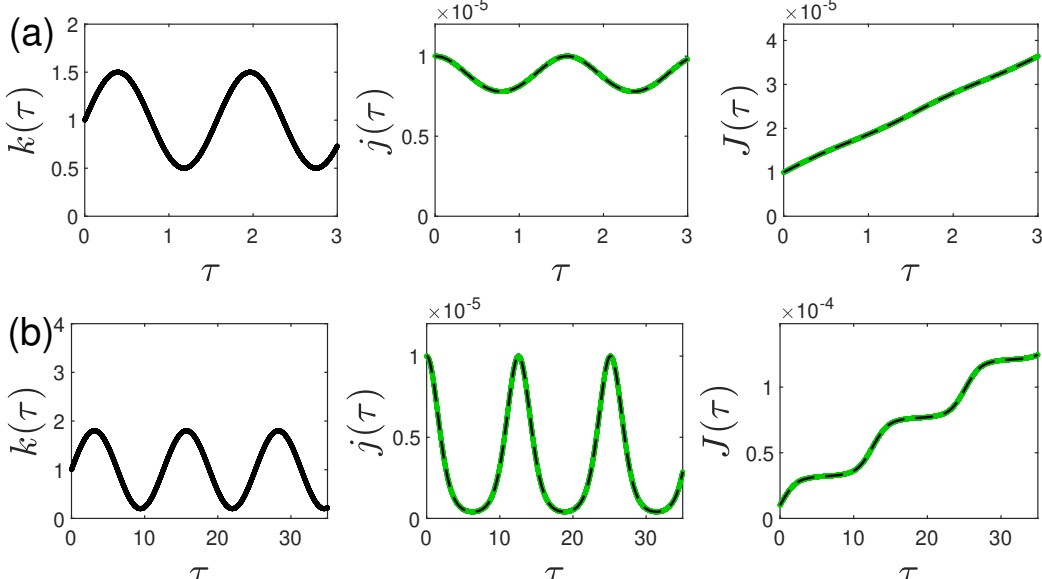

**Figure 3.** Left panels: Plot of the oscillating ratio of $k(\tau)$ according to Equation (46) for (**a**) $\alpha = 0.5$, $\beta = 4$, and (**b**) $\alpha = 0.8$, $\beta = 0.5$. Centered panels: The corresponding rates of new infections of $j(\tau)$ as a function of reduced time using $\eta = 10^{-5}$. Right panels: Cumulative fraction of $J(\tau)$. Shown are the numerical (black dashed curve) solutions of the SIR set of equations in (6) in comparison with the analytical approximations (green curve) according to Equations (47) and (48). Because $\alpha/\beta \ll 1$ for case (**a**), only the first term of expansion (48) had to be used, while the first three terms of Equation (48) were used in (**b**) in accord with Figure 4. The agreement is almost perfect. The maximum relative deviations are smaller than 0.2% for all cases.

Integrating the rate of (47) over reduced time provides us with the corresponding cumulative fraction,

$$
\begin{aligned}
J(\tau \leq \tau_c) &= \eta + \frac{\eta(1-\eta)}{\beta} e^{-\frac{\alpha}{\beta}} \int_0^{\beta\tau} dy\, e^{\frac{\alpha}{\beta}\cos(y)} \\
&= \eta + \eta(1-\eta) e^{-\frac{\alpha}{\beta}} \left[ \tau I_0\left(\frac{\alpha}{\beta}\right) + 2 \sum_{n=1}^{\infty} \frac{I_n\left(\frac{\alpha}{\beta}\right)}{n\beta} \sin(n\beta\tau) \right],
\end{aligned}
\tag{48}
$$

where we have used $J(0) = \eta$ and the series expansion (Equation 9.6.34 of ref. [58])

$$
e^{z\cos\theta} = I_0(z) + 2 \sum_{n=1}^{\infty} I_n(z)\cos(n\theta)
\tag{49}
$$

in terms of the modified Bessel function of the first kind $I_n(z)$. In order to obtain deviations of less than one percent from the series (49) at a finite summation index $n$, one has to choose $N$ according to Figure 4. For the example provided in Figure 3a, $z = \alpha/\beta = 0.5/4 \ll 1$, just the first term of the expansion, $N = 1$, is sufficient to capture the behavior of $J(\tau)$ for this case. Instead, $N = 3$ is used to calculate $J(\tau)$ in Figure 3b, which is in accord with the value of $z = 0.8/0.5 = 1.6$ in Figure 4.

The third panel of Figure 3 and Equation (48) show that the cumulative fractions predominantly increase linearly with reduced time so that at some finite time $\tau_c$, the cumulative fractions approach $J_\infty$. There, the validity of the early time approximation ends and $J(\tau \geq \tau_c) \simeq J_\infty$, as is discussed in Appendix B. The value of $\tau_c$ and the variation in the corresponding late-time spontaneous rate $j(\tau \geq \tau_c)$ can be calculated according to Equations (A15) and (A20), respectively.

From the part of Equation (48) that is proportional to $\tau$, one can read off the characteristic time $\tau_c$ using $J(\tau_c) \simeq J_\infty$, which translates to

$$
\tau_c \simeq \frac{e^{\alpha/\beta}}{I_0\left(\frac{\alpha}{\beta}\right)} \frac{J_\infty}{\eta(1-\eta)}.
\tag{50}
$$

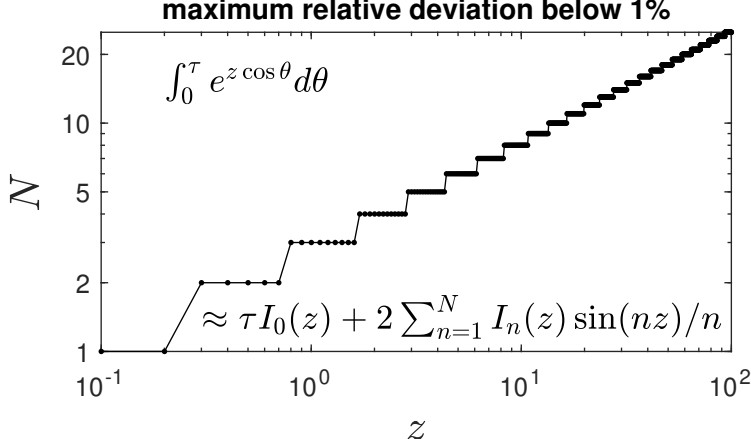

**Figure 4.** The integral $\int_0^\tau e^{z\cos\theta}$ is approximated for all $\tau$ within 1% precision by the integrated Equation (49), thus resulting in the expression shown inside the figure if the $z$-dependent order of the summation, $N$, is chosen as depicted. The required order grows as $N \propto \sqrt{a}$.

## 6. Summary and Conclusions

The dynamical equations of the SIR epidemics model play an important role in predicting and/or analyzing the temporal evolution of epidemic outbreaks. Crucial input quantities are the time-dependent infection ($a(t)$) and recovery ($\mu(t)$) rates, which regulate the transitions between the compartments $S \to I$ and $I \to R$, respectively. Accurate analytical approximations for the temporal dependence of the rate of new infections $\mathring{J}(t) = a(t)S(t)I(t)$ and the corresponding cumulative fraction of infections $J(t) = J(t_0) + \int_{t_0}^{t} dx \mathring{J}(x)$ are available in the literature for either stationary infection and recovery rates or for a stationary value of the ratio $k(t) = \mu(t)/a(t)$. Here apparently for the first time, a new accurate analytical approximation has been derived for arbitrary and different temporal dependencies of the infection and recovery rates, which is valid for not-too-late times after the start of the infection when the cumulative fraction $J(t)$ is much less than unity. Equations (12) and (15) provide analytical expressions for the rate of new infections as a function of real time $j(t)$ and reduced time $j(\tau)$, with the reduced time $\tau = \int_{t_0}^{t} dx a(x)$. Likewise, Equation (14) gives the corresponding cumulative fraction $J(t) = J(\tau)$.

The comparison of the analytical approximation with the exact numerical solution of the SIR equations for different illustrative examples proves the accuray of the analytical approach. These examples include the cases of a monotonically rising ratio, as well as an oscillating ratio as a function of reduced time. The former one is well suited to represent a single wave of a pandemic outburst, where, during the outburst, the recovery rate becomes greater than the infection rate due to the combined effects of nonpharmaceutical interventions and/or the dedicated medication or vaccination of infected people or noninfected persons, respectively. The case of an oscillating ratio is well suited to represent a series of repeating epidemic outbursts over a longer time span. The future predictions or analyses of epidemics on the basis of the SIR model equations will certainly benefit from the newly derived analytical solutions.

**Author Contributions:** Conceptualization, R.S.; methodology, R.S. and M.K.; software, M.K., writing—review and editing, R.S. and M.K. All authors have read and agreed to the published version of the manuscript.

**Funding:** This research received no external funding.

**Institutional Review Board Statement:** Not applicable.

**Informed Consent Statement:** Not applicable.

**Data Availability Statement:** The data are contained within the article.

**Conflicts of Interest:** The authors declare no conflict of interest.

## Appendix A. Cumulative Fraction for General Reduced Time Dependencies $k(\tau)$

In the examples discussed in Sections 4 and 5, the cumulative fractions have been calculated by integrating the respective rates of new infections. If such exact integrations are not possible for general reduced time dependencies of the ratio $k(\tau)$, we suggest to use the method of steepest descent [59,60]. Here, we write the cumulative fraction (13) as

$$J(\tau \leq \tau_c) = \eta + \eta(1 - \eta)F(\tau), \tag{A1}$$

with

$$F(\tau) = \int_0^\tau dx\, e^{-f(x)}, \qquad f(x) = \int_0^x d\xi\, k(\xi) - x. \tag{A2}$$

We expand the function $f(x)$ to the second order near its minima values at $x = \tau_m$, which are given by

$$k(\tau_m) = 1. \tag{A3}$$

We emphasize that, depending on the choosen variation of the ratio $k(\tau)$, there may be several minima, as in the case of the oscillating ratio discussed above in Section 5.2. Then, we have

$$f(x) \simeq f(\tau_m) + \frac{1}{2}k'(\tau_m)(x - \tau_m)^2. \tag{A4}$$

For a minimum to occur, the ratio has to have an increasing reduced time variation with $k'(\tau_m) > 0$. With the approximation (A4), the integral (A2) can be evaluated in terms of error functions, thus providing

$$
\begin{aligned}
F(\tau) \quad \simeq \quad & \sum_m \sqrt{\frac{\pi}{2k'(\tau_m)}} \exp\left(\tau_m - \int_0^{\tau_m} d\xi\, k(\xi)\right) \times \\
& \left[\operatorname{erf}\left(\sqrt{\frac{k'(\tau_m)}{2}}(\tau - \tau_m)\right) + \operatorname{erf}\left(\sqrt{\frac{k'(\tau_m)}{2}}\tau_m\right)\right].
\end{aligned}
\tag{A5}
$$

Consequently, the cumulative number of infections (A1) becomes Equation (14). In principle, calculating the cumulative fraction by the method of steepest descent is essentially identical to approximating the corresponding rate of new infections using a sum of Gaussian distributions centered at their maxima. If there is only one maximum, as in the case discussed in Section 5.1, the method of steepest descent provides a lower limit to the cumulative fraction, because all other contributions far from the maximum are not adequately accounted for.

## Appendix B. Solution at Late Times

*Appendix B.1. Limit $J \simeq J_\infty = 0.7$*

In this limit, $R_\infty = J_\infty$, $S_\infty = 1 - J_\infty = 0.3$, $I_\infty = 0$, and $j(\tau = \infty) = 0$. For values of $k \gg S_\infty = 0.3$, we approximate Equation (10) as

$$k(\tau) \simeq -\frac{d \ln I(\tau)}{d\tau}. \tag{A6}$$

Equation (A6) integrates to

$$I(\tau \geq \tau_c) \simeq A_1 e^{-\int_0^\tau d\xi\, k(\xi)}, \tag{A7}$$

with the integration constant being $A_1$. The approximation (A7) is in agreement with the final value of $I_\infty = 0$. Integrating Equation (6c) with Equation (A8) inserted then readily provides

$$R(\tau \geq \tau_c) \simeq A_2 - A_1 e^{-\int_0^\tau d\xi\, k(\xi)} = A_2 - I(\tau > \tau_c) \tag{A8}$$

with the further integration constant being $A_2$, which has to be set to $A_2 = J_\infty$ so that

$$R(\tau \geq \tau_c) \simeq J_\infty - A_1 e^{-\int_0^\tau d\xi\, k(\xi)}, \tag{A9}$$

thereby implying that

$$
\begin{aligned}
J(\tau \geq \tau_c) \quad &= \quad 1 - S(\tau \geq \tau_c) = I(\tau \geq \tau_c) + R(\tau \geq \tau_c) = J_\infty, & \tag{A10}\\
S(\tau \geq \tau_c) \quad &= \quad 1 - J_\infty, & \tag{A11}\\
j(\tau \geq \tau_c) \quad &= \quad S(\tau \geq \tau_c)I(\tau \geq \tau_c) = (1 - J_\infty)A_1 e^{-\int_0^\tau d\xi\, k(\xi)}. & \tag{A12}
\end{aligned}
$$

In terms of the real time in this late-time limit, we have

$$
\begin{aligned}
J(t \geq t_c) \quad &= \quad J_\infty = 0.7, & \tag{A13}\\
\mathring{J}(t \geq t_c) \quad &= \quad a(t)(1 - J_\infty)A_1 e^{-\int_{t_0}^t dx\, \mu(x)}. & \tag{A14}
\end{aligned}
$$

Due to our approximation of the exact SIR equations, the solutions (A10) and (A12), as well as (A13) and (A14), no longer fulfill the properties $j(\tau) = dJ(\tau)/d\tau$ and $\mathring{J}(t) = dJ(t)/dt$. We regard this inconsistency as minor and therefore tolerable, because the exponentially decaying rates of (A12) and (A14) of new infections contribute little to the corresponding cumulative fractions. The constant $A_1$ appearing in the approximate solutions of (A7), (A9), (A12), and (A14) is determined by the continuity conditions of the rate of new infections and the cumulative fraction at $\tau_c$, with the approximate solutions for (12)–(13) existing at small values of $J \ll 1$.

*Appendix B.2. Continuity Conditions*

The reduced time of $\tau_c$ is determined by equating the early (13) and late (A10) cumulative number approximations at $\tau_c$ to yield

$$\int_0^{\tau_c} dx\, e^{x - \int_0^x d\xi k(\xi)} = \frac{J_\infty}{\eta(1-\eta)} = \frac{0.7}{\eta(1-\eta)}. \tag{A15}$$

For prescribed reduced time variations of $k(\tau)$, Equation (A15) can be reduced further.

Likewise, with the accordingly determined value of $\tau_c$, the constant $A_1$ is determined by equating the early (12) and late (A12) rate of new infections at $\tau_c$ to yield

$$A_1 = \frac{\eta(1-\eta)}{1-J_\infty} e^{\tau_c}. \tag{A16}$$

Consequently, with (A16), the late time approximations (A7), (A9), (A11), and (A12) become

$$I(\tau \geq \tau_c) = \frac{\eta(1-\eta)}{1-J_\infty} e^{\tau_c - \int_0^\tau d\xi k(\xi)}, \tag{A17}$$

$$R(\tau \geq \tau_c) = J_\infty - \frac{\eta(1-\eta)}{1-J_\infty} e^{\tau_c - \int_0^\tau d\xi k(\xi)}, \tag{A18}$$

$$S(\tau \geq \tau_c) = 1 - J_\infty, \tag{A19}$$

$$j(\tau \geq \tau_c) = \eta(1-\eta) e^{\tau_c - \int_0^\tau d\xi k(\xi)}. \tag{A20}$$

**Appendix C. Expansion of $H(\tau)$ at Small Times**

Substituting $x = 1 + z$ into Equation (35) provides for the integral of (A15):

$$H(\tau) = \frac{2^{B/C}}{2C} \int_0^{e^{2C\tau}-1} dz\, (1+z)^{\frac{B+1}{2C}-1} (2+z)^{-B/C}$$

$$= \frac{1}{2C} \int_0^{e^{2C\tau}-1} dz\, (1+z)^{\frac{B+1}{2C}-1} \left(1+\frac{z}{2}\right)^{-B/C}. \tag{A21}$$

For times that are much smaller than $\tau \ll \ln 2/(2C)$, one notices that $e^{2C\tau} - 1 \ll 1$, i.e., $e^{2C\tau} - 1 \approx 2C\tau + (2C\tau)^2/2 + (2C\tau)^3/6 \equiv Z$. Expanding the integrand for small values of $z \leq e^{2C\tau} - 1 \ll 1$ to the second order in $z$ provides the approximation

$$H\left(\tau \ll \frac{\ln 2}{2C}\right) \simeq \frac{1}{2C} \int_0^Z dz \left[1 + \frac{1-2C}{2C}z + \left(1 - \frac{6+B}{8C} + \frac{1}{8C^2}\right)z^2\right]$$

$$\simeq \tau + \frac{\tau^2}{2} + \frac{(1-BC)}{6}\tau^3 + \mathcal{O}(C\tau)^4, \tag{A22}$$

which is the expression mentioned after Equation (36).

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
