# Peer review of "Analytical Solution of the Susceptible-Infected-Recovered/Removed Model for the Not-Too-Late Temporal Evolution of Epidemics for General Time-Dependent Recovery and Infection Rates"

_covid, doi:10.3390/covid3120123_

Round 1

Reviewer 1 Report

Comments and Suggestions for Authors

Overall, this an interesting paper and well presented. The paper should be accepted for publication with some minor revisions.

Line 15 : I personally would write the model as "susceptible-infected-recovered", not "recovered/removed"

Line 19 : is very hard to read the dot over the J. I would use $\frac{dJ}{dt}$

Line 20 : Another personal preference, but please write integrals as $\int_{t_0}^t J(x)\, dx$ not $\int_{t_0}^t dx J(x)$

Line 23 : you have "J(t) << 1" and "is much less than unity". Use either one or the other, not both.

Line 41 - remove "also" and "probably"

Line 56 - I do not know what you mean by the term "semi-time". Should this be "semi-infinite"?

Line 74 - has t_0 been set to zero?

Line 76 - I think the first line of (10) should be $\ln S(\tau)^{-1}$ - it should not be the inverse function of $S(\tau)$!

Equation 27 - it should be $arctan$ not $artan$

Equation 30 - there should be an equal sign on the second line.

Figure 2 and 3 - make the plots as two columns of three rows, rather than two rows of three columns. It lets the figures be larger (and readible)

Line 196 - remove "positive"

Figure 3 caption - replace "ist" with "is"

There are a number of corrections mentioned above which are repeated throughout the manuscript.

Finally, I personally do not seen any value by including the appendices.

Comments on the Quality of English Language

Some minor changes are needed.

Author Response

REFEREE: Overall, this an interesting paper and well presented. The paper should be accepted for publication with some minor revisions.

AUTHORS: We thank this referee for a careful reading and positive assessment. 

REFEREE: Line 15 : I personally would write the model as "susceptible-infected-recovered", not "recovered/removed"

AUTHORS: We think recovered/removed is correct as the model absorbs the deceased fraction into the R compartment. Since this phrase does only appear once in the manuscript, we did not change it. 

REFEREE: Line 19 : is very hard to read the dot over the J. I would use $\frac{dJ}{dt}$

AUTHORS: We fully agree and changed the symbol (all changes highlighted in the revised manuscript).

REFEREE: Line 20 : Another personal preference, but please write integrals as $\int_{t_0}^t J(x)\, dx$ not $\int_{t_0}^t dx J(x)$

AUTHORS: We kept the present version to avoid any mistakes. 

REFEREE: Line 23 : you have $J(t) << 1$ and "is much less than unity". Use either one or the other, not both.

AUTHORS: Thanks, we adjusted this sentence. 

REFEREE: Line 41 - remove "also" and "probably"

AUTHORS: We removed 'also' and kept 'probably'.

REFEREE: Line 56 - I do not know what you mean by the term "semi-time". Should this be "semi-infinite"?

AUTHORS: Thanks for this important comment. We now explain 'semi-time' in the revised text. 

REFEREE: Line 74 - has $t_0$ been set to zero?

AUTHORS: No, note that tau(t0)=0, this is now also mentioned in the revised text. 

REFEREE: Line 76 - I think the first line of (10) should be $\ln S(\tau)^{-1}$ - it should not be the inverse function of $S(\tau)$!

AUTHORS: We agree, and did the change. 

REFEREE: Equation 27 - it should be $arctan$ not $artan$

AUTHORS: We now switched to artanh, while both are in use. 

REFEREE: Equation 30 - there should be an equal sign on the second line.

AUTHORS: Thanks, corrected! 

REFEREE: Figure 2 and 3 - make the plots as two columns of three rows, rather than two rows of three columns. It lets the figures be larger (and readible)

AUTHORS: Rows 1 and 2 are different cases, we therefore prefer to keep the ordering. 

REFEREE: Line 196 - remove "positive"

AUTHORS: We consider only cases with positive \alpha and \beta and therefore prefer to explicitly mention this.

REFEREE: Figure 3 caption - replace "ist" with "is"

AUTHORS: done, here and in other places.

REFEREE: There are a number of corrections mentioned above which are repeated throughout the manuscript.

AUTHORS: We corrected similar typos in other places. 

REFEREE: Finally, I personally do not seen any value by including the appendices.

AUTHORS. All appendices are mentioned in the main text, some of them provide proofs for equations used in the main text. We need for example Appendix B to construct the late time dependence referred to in Eq. (42).  Thats why we need to keep the appendices. 

Reviewer 2 Report

Comments and Suggestions for Authors

The authors presented an analysis of the susceptible-infected-recovered/removed (SIR) epidemics model. Namely, they obtained approximate analytical solutions of SIR equations. The paper assumes that the cumulative number of new cases of infection is much less than 1 (a case of not too late times). Besides, there was also analyzed the case of late times, when the fraction of susceptible persons is much smaller than 1.

The authors show that, within the illustrative examples presented in their paper, the analytical approximations perfectly correspond to the results of numerical integration of the SIR model equations. Specifically, the authors analyzed both the case of a wave of a pandemic outburst when the recovery rate exceeds the infection rate, and the case of a series of repeating epidemic outbursts. Both of these cases can be of interest to researchers of the spread of infections

The authors state that “Data sharing is not applicable to this article”. However, in a broader context, the analysis of epidemics, as a rule, requires the involvement of reliable data. The reliable data, for example, S(t), I(t) and R(t), can be displayed as time series. If this is the case, then the functions a(t) and μ(t) could easily be obtained also in the form of time series specific to a particular infectious process. This approach (the KDD approach, which provides for the direct inclusion of time series in the equations) has been proposed in Medvinsky et al. (2023) Sci. Rep.13, 10124.  In this regard, I have the following question. Would the authors like to include in their paper a discussion of the prospects of using the KDD approach for the analysis of infections in comparison with the analytical solutions of the equations of the SIR model? I, as well as possibly other readers, would be interested to know the opinion of the authors on this issue.

Regardless of the authors' answer to my question, I recommend this paper for publication.

Author Response

We thank this referee for a careful reading and positive assessment. 

REFEREE: The authors state that “Data sharing is not applicable to this article”. However, in a broader context, the analysis of epidemics, as a rule, requires the involvement of reliable data. The reliable data, for example, S(t), I(t) and R(t), can be displayed as time series. 

AUTHORS: We do not agree. Reliable quantities S(t), I(t) and R(t) have never been reported. The only reliably reported quantity is J(t) and its time-derivative.  

REFEREE: If this is the case, then the functions $a(t)$ and $\mu (t)$ could easily be obtained also in the form of time series specific to a particular infectious process. This approach (the KDD approach, which provides for the direct inclusion of time series in the equations) has been proposed in Medvinsky et al. (2023) Sci. Rep.13, 10124.  In this regard, I have the following question. Would the authors like to include in their paper a discussion of the prospects of using the KDD approach for the analysis of infections in comparison with the analytical solutions of the equations of the SIR model? I, as well as possibly other readers, would be interested to know the opinion of the authors on this issue.

AUTHORS: We did in Ref (49) something similar, and determined k(t) from the above-mentioned observable. We are not planning to apply the interesting KDD approach to the present work, and therefore prefer to leave this to others.